# SELECT VIA PROXY: EFFICIENT DATA SELECTION FOR TRAINING DEEP NETWORKS

## ABSTRACT

At internet scale, applications collect a tremendous amount of data by logging user events, analyzing text, and collecting images. This data powers a variety of machine learning models for tasks such as image classification, language modeling, content recommendation, and advertising. However, training large models over all available data can be computationally expensive, creating a bottleneck in the development of new machine learning models. In this work, we develop a novel approach to efficiently select a subset of training data to achieve faster training with no loss in model predictive performance. In our approach, we first train a small proxy model quickly, which we then use to estimate the utility of individual training data points, and then select the most informative ones for training the large target model. Extensive experiments show that our approach leads to a $1.6\times$ and $1.8\times$ speed-up on CIFAR10 and SVHN by selecting 60% and 50% subsets of the data, while maintaining the predictive performance of the model trained on the entire dataset.

## 1 INTRODUCTION

Large-scale labeled data has been critical to the recent success of deep learning (Halevy et al., 2009; Sun et al., 2017; Hestness et al., 2017) and there are a variety of domains and settings where labeled data is plentiful, such as predicting the next word or character in language modeling and predicting user-provided image tags for image classification. However, for these large datasets, training deep networks can incur prohibitively long training times, measured in days, weeks, or even months (Sun et al., 2017). This overhead impedes the development of new machine learning models and uses large amounts of computational resources (Amodei & Hernandez, 2018).

Subsampling training data is a common solution to this problem, but naive, uniform subsampling can miss important rare examples. For instance, in many web applications, data is abundant for a small subset of core users/content but is scarce for new users/content. Similarly, when training autonomous vehicles, yellow lights occur less frequently than green and red lights but are equally important (Karpathy, 2018). While more sophisticated methods such as core-set selection techniques can select a representative subset, these methods are either algorithm-specific (Har-Peled & Kushal, 2007; Tsang et al., 2005; Huggins et al., 2016; Campbell & Broderick, 2017; 2018) and don't apply to deep learning, or require a meaningful feature representation (Wei et al., 2013; 2014; Tschiatschek et al., 2014; Ni et al., 2015) that must either be hand-designed or learned for unstructured data, which requires substantial computational resources.

In this paper, we introduce a new method called *Select Via Proxy (SVP)* that provides a computationally efficient way of training large deep learning models while empirically maintaining model quality/performance. Our key idea is to quickly select a subset of the most informative data and then train a large model on this subset. Training with less data leads to faster training times, while training on the most informative data allows the model to learn as well as if it were trained on the full data. To identify informative data points we first quickly train a small proxy model, and then use this proxy to select data for training the large full model. The proxy model is a simple and fast-to-train model—for example, a small model architecture (e.g., ResNet20 instead of ResNet164) trained for a few epochs (e.g., 25% of the desired epochs). We subsequently select a training set for the larger model based on the predictions generated by this smaller proxy model. This selection process can be thought of as a preprocessing step that filters data before feeding it into an existing training pipeline,

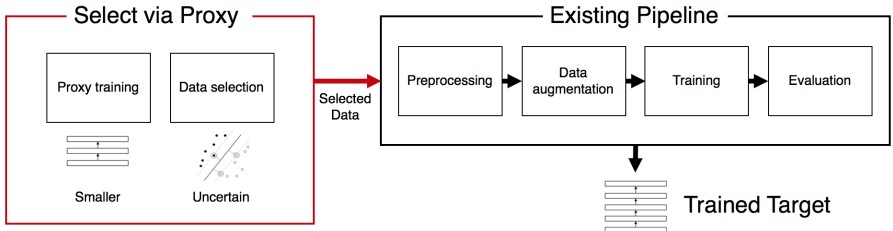

Figure 1: Select Via Proxy (SVP) can be viewed as an additional preprocessing step to an existing training pipeline. A small proxy model is trained over all the data available, but only a selected subset are used to train the large target model. The training procedure of the target model is not changed, lowering the overhead of implementing this method.

as depicted in Figure 1. To select the points with highest informativeness, we leverage uncertainty sampling (Lewis & Gale, 1994) from active learning, where uncertainty can be measured by metrics such as entropy of the output probabilities. However, in active learning a model is generally trained to select the next point (or batch) (Settles, 2012), which is efficient in terms of labels, but often computationally expensive. While this can be effective when deciding which data to acquire labels for from an expensive labeler (e.g. a human), the computational cost is too high to accelerate training over an existing large labeled dataset. Using a proxy reduces the cost of selection by up to a $100\times$.

The main benefits of our proposed method are (i) it can be easily added to a training pipeline without modifying the training procedure of the target model and (ii) the proxy is very fast to train and can substantially improve the training time of large deep models while maintaining the predictive performance. Extensive experimental evaluation shows the effectiveness of our approach. In general, the proxy is very fast to train. As a result, including the time to train the proxy, select points using the trained proxy, and train the large target model on the selected points, we find that SVP can substantially reduce training time compared to training over all the data while maintaining the same predictive performance. SVP leads to a $1.6\times$ and $1.8\times$ speed-up on CIFAR10 (Krizhevsky & Hinton, 2009) and SVHN (Netzer et al., 2011) by selecting 60% and 50% subsets of the data respectively while maintaining predictive performance. To explore the robustness of our proposed framework to the choice of proxy model architecture, we combine two approaches for sentiment analysis that represent separate extremes in speed and accuracy. Our experiments show that SVP with fastText (Joulin et al., 2016) as a proxy model can be used to remove 20% of the data from Amazon Review Polarity (Zhang et al., 2015; He & McAuley, 2016) for the target model VD-CNN29 (Conneau et al., 2017), a fundamentally different architecture from the proxy that is over $100\times$ slower to train (e.g., 17 hours instead of 10 minutes). This provides a simple but effective means of decreasing training time without increasing error.

## 2  RELATED WORK

SVP builds upon a large body of related work on sampling and training data selection.

**Core-set selection.** Core-set selection attempts to find a representative subset of points to speed up learning or clustering; such as $k$-means and $k$-medians (Har-Peled & Kushal, 2007), SVM (Tsang et al., 2005), Bayesian logistic regression (Huggins et al., 2016), and Bayesian inference (Campbell & Broderick, 2017; 2018). However, these examples are algorithm-specific and do not directly apply to deep neural networks. Very recently, Sener & Savarese (2018) used core-set selection for active learning with convolutional neural networks. While this technique reduces sample complexity, the proposed technique is computationally intensive and does not save on training time. There is a body of work on data summarization based on submodular maximization (Wei et al., 2013; 2014; Tschiatschek et al., 2014; Ni et al., 2015). However, the techniques here are rather different as they do not incorporate a model prediction.

**Subset selection to increase accuracy.** Recently, Chang et al. (2017) proposed to choose data points whose predictions have changed most over the previous epochs as a lightweight estimate of uncertainty. From the machine teaching literature, Fan et al. (2018) demonstrated that data selection can be learned through reinforcement learning. By repeatedly training a student model over the

dataset or similar dataset, they train a teacher model to filter points for each update of the student model using the student's final accuracy and training time as a reward. In comparison to our method, both of these approaches have substantial burn in periods (i.e., building a history of predictions or training the teacher) before data can be effectively selected.

In addition, Wang et al. (2018b) trains a model, selects a subset, and then retrains the same model on a subset of the data where a small fraction of unfavorable training examples ($< 5\%$) are removed to give higher accuracy. The goal of the above work is to increase the accuracy of training rather than speed up training. However, these results make it less surprising that we can achieve the same accuracy with a subset of the full dataset even though we are able to remove up to $12\times$ more points.

**Heterogeneous active learning.** In the active learning literature, there are examples of using one model to select points for a different, more expensive model. For instance, (Lewis & Catlett, 1994) uses a probabilistic classifier to select points to label for a decision tree target model. Tomanek et al. (2007) uses a committee-based active learning algorithm for an NLP task and notes that the set of selected points are "reusable" across different models (maximum entropy, conditional random field, naive Bayes). In light of this work, we demonstrate this phenomena generalizes to modern deep learning models where model capacity and training time can be substantially reduced to effectively select informative examples.

**Optimization and Importance Sampling.** There is a large literature on weighting or sampling datapoints in the optimization procedure in order to speed up training or achieve higher accuracy. The most common paradigm is to perform importance sampling on training data points based on the gradient norm (Alain et al., 2015), loss (Loshchilov & Hutter, 2015), bound on the gradient norm (Katharopoulos & Fleuret, 2017), or approximate loss (Katharopoulos & Fleuret, 2018). There has also been work on learning the weights or sampling probabilities to minimize the variance of the stochastic gradient (Bouchard et al., 2015; Borsos et al., 2018; Gopal, 2016). In the theoretical optimization literature, there is work on importance sampling data points for faster convergence (Needell et al., 2014; Zhao & Zhang, 2015; Allen-Zhu et al., 2016). There has also been a line of work in the Neural Machine Translation literature focused on reducing training time by focusing more on "hard" examples in later epochs (Zhang et al., 2017; van der Wees et al., 2017; Wang et al., 2018a; Kocmi & Bojar, 2017). In comparison, our approach does not modify the training procedure of the target model, making it an easy addition to an existing training pipeline.

We note that our proposed method is orthogonal to the above. SVP could be used in conjunction with the algorithms above to further decrease training time or reduce computational complexity in the case of active learning and core-set selection, but we leave this for future work.

## 3 SELECT VIA PROXY

We now present our main contribution, Select via Proxy (SVP), for efficiently training expensive deep learning models. Our approach consists of three steps: 1) Create a proxy model that is fast to train and to provide us with an approximate decision boundary, 2) use the proxy model to select a subset of uncertain data points around the decision boundary, and 3) train the large target model on the selected subset via proxy to refine the decision boundary and get the final accurate model. In the following section, we discuss each step in detail.

### 3.1 CREATING A PROXY MODEL

The key idea behind the proxy model is to create a small model that is fast to train and can provide a good approximation of the decision boundary of the large target model. In order to create such a model we rely on the following observations.

**Creating a proxy by scaling down the target model.** It has been observed that for deep models with many layers, reducing the dimension (narrowing) or number of hidden layers (shortening) leads to a considerably reduced training times with only a small drop in accuracy. For example, in image classification, the accuracy of deep networks with residual connections (e.g., ResNet164 and ResNet110) only slightly diminishes as layers are dropped from the network (He et al., 2016b;a). As Figure 2a shows, a model with 20 layers achieves an accuracy of 92.1% in 22 minutes, while a

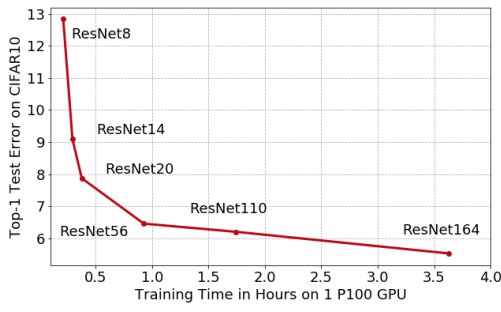 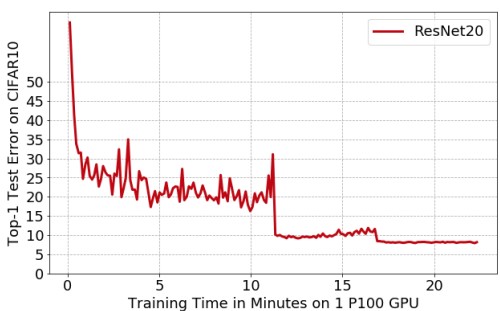

(a) Top-1 test error and training time on CIFAR10 for ResNet with pre-activation and a varying number of layers. There is a diminishing returns in accuracy by increasing the number of layers.

(b) Top-1 test error during training of ResNet20 with pre-activation. In the first 12 minutes, ResNet20 reaches 9.2% top-1 error, while the remaining 10 minutes are spent on increasing accuracy to 7.9%

Figure 2: Top-1 test error on CIFAR10 for varying model sizes (left) and over the course training a single model (right), demonstrating a large amount of time is spent on small changes in accuracy.

larger model with 164 layers only improves performance by 2.4%, but takes 3 hours and 40 minutes to train.

Similar results have been shown for scaling down networks with a variety of model architectures (Huang et al., 2016; Xie et al., 2017; Huang et al., 2017) and a number of other tasks including language modeling, neural machine translation, text classification, and recommendation (Conneau et al., 2016; He et al., 2017; Jozefowicz et al., 2016; Dauphin et al., 2017; Vaswani et al., 2017). We exploit the diminishing returns property between training time and reductions in error to scale down a given target model to a small proxy that can be trained quickly but still provides a good approximation of the decision boundary of the target model.

**Training for a smaller number of epochs.** As shown in Figure 2b, a significant amount of training is spent to obtain a relatively small reduction in error. While training ResNet20, almost half of the training time (i.e., 10 minutes out of 22 minutes) is spent on a 1.3% improvement in error. Based on the above observation, we can train the proxy model for a smaller number of epochs and still get a good approximation from the decision boundary of the target model.

**Boosting the performance by ensembling small models.** Boosting is a common approach to get a strong learner by ensembling a set of weak learners. Similarly, a stronger proxy can be obtained by training a set of small models and combining their predictions. The stronger proxy can provide a better approximation from the decision boundary of the target model. Considering that the much smaller size of the proxy model compared to the size of the target model, multiple proxy models can be trained with little or no additional resources in parallel, with no increase to overall training time. Each learner within the ensemble has its own noisy approximation of the decision boundary. Hence, the small learners can be combined together through *alternating selection* or *rank combination* (Settles, 2011; Reichart et al., 2008). Figure 3 shows that ensembling improves the already high correlation between the proxy model's ranking of points based on entropy and the same ranking produced by the model trained on the whole data.

## 3.2 SUBSET SELECTION VIA PROXY

Having trained the small proxy model on the entire dataset, we can use its predictions to select informative subsets to train the large target model. As discussed in Section 3.1, the proxy model provides an approximation from the decision boundary between different classes. Considering that the large target model is able to learn a more refined decision boundary, we use the proxy model to select the most uncertain data points around the decision boundary (Lewis & Gale, 1994; Lewis & Catlett, 1994) and subsequently train the target model on the uncertain subsets.

**Quantifying uncertainty.** Various uncertainty metrics can be used, including *confidence*, *margin*, and *entropy* (Settles, 2012). For a classifier that for every data point $x$ provides the probability

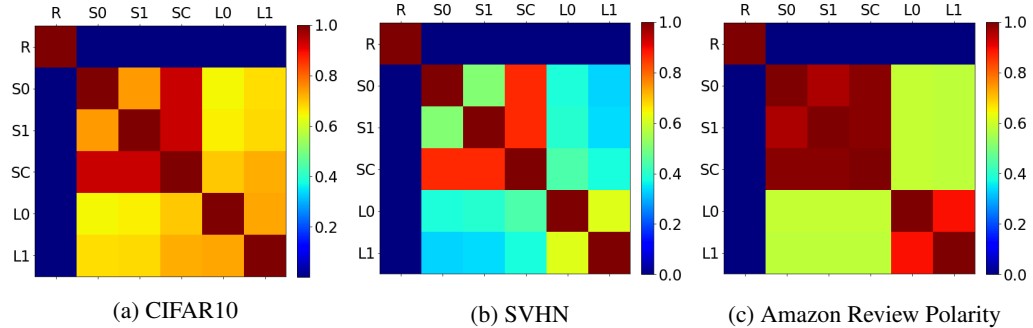

(a) CIFAR10        (b) SVHN        (c) Amazon Review Polarity

Figure 3: Pearson product-moment correlation of examples ranked by entropy calculated from different models on CIFAR10 using ResNet20 and ResNet164 with pre-activation (left), SVHN using ResNet20 and ResNet152 (center), and Amazon Review Polarity using fastText and VDCNN29 (right). S1 and S2 represent two separate runs of the small proxy model (e.g., ResNet20), while L1 and L2 represent different runs of the large target model (e.g., ResNet164). R gives a random order of points for reference. On all datasets, ensembling multiple small models together through rank combination (SC) increases the Pearson product-moment correlation with the large model.

---

**Algorithm 1** SELECT VIA PROXY (SVP)

---

**Input:** Data set $D$, cardinality $k$, deep model architecture $\mathcal{M}$.
**Output:** Trained deep model $\mathcal{M}^t$.
 1: Create a proxy model by scaling down the target model as described in section 3.1.
 2: Train the small proxy model on the entire dataset $D$.
 3: Calculate uncertainty of data points via the proxy model using uncertainty metrics from section 3.2.
 4: Sort the examples in a decreasing order based on their uncertainty.
 5: Train the target model $\mathcal{M}$ on the subset $S$ of top $k$ uncertain examples to get the final output $\mathcal{M}^t$.
 6: **return** $\mathcal{M}^t$.

---

$P(y|x)$ for $x$ to belong to class $y$, the uncertainty function $f$ can be defined as follow.

$$f_{\text{confidence}}(x) = 1 - P(\hat{y}|x) \tag{1}$$
$$f_{\text{margin}}(x) = 1 - \min_{y \neq \hat{y}}(P(\hat{y}|x) - P(y|x)) \tag{2}$$
$$f_{\text{entropy}}(x) = -\sum_{y} P(y|x) \log P(y|x), \tag{3}$$

where $\hat{y} = \arg\max_y P(y|x)$ is the most probable label for data point $x$. For all the above metrics, a value of 0 means no uncertainty and higher values mean more uncertainty. In general, we usually care about the ordering of data points by each uncertainty measure and not the uncertainty values. Note that for binary labels, the ordering between these three metrics are equivalent. The set of most uncertain $k$ points in dataset $D$, i.e. $S = \arg\max_{A \subseteq D : |A| \leq k} \sum_i f(x_i)$ can then be obtained by sorting the data points according to each uncertainty metric, and taking the top $k$ uncertain points in the sorted order.

**Training the target model on the subsets selected via proxy.** Finally, the set of uncertain data points can be used to train the large target model. Here, the idea is to refine the decision boundary learned by the proxy model. Since the target model is able to learn a more complex decision boundary, we select the data points around the approximate decision boundary learned by the proxy and let the target model refine the decision boundary of the proxy model. The Pseudocode of the proposed method is outlined in Alg. 1.

**Choosing subset cardinality.** While our algorithm assumes the subset size is given, we describe two important use cases where this is a reasonable assumption. First, for large-scale applications where data is constantly collected based on user interactions, models are often re-trained periodically on the most recent data. In this case, the target model class is known, and the subset size can be determined once from historical data. Second, the dataset is too large to train on, forcing the data to

be subsampled to a fixed size to meet a computational or time budget. In this case, the subset size is known, and a new proxy can be created for each target. As demonstrated in section 4, in most cases, our method performs better than random for a fixed subset size with little additional overhead.

We note that while we discussed different steps in Section 3.1 to create the proxy model, and explored various uncertainty measures in Section 3.2 to select data points via proxy, our method is robust to a wide range of model architectures, training routines, and metrics for creating the proxy and selecting data from it. In section 4, we demonstrate this robustness concretely through a series of experiments examining our choice of proxy model relative to a given target model.

## 4  RESULTS

To investigate the performance of SVP (Alg. 1), we perform experiments on three datasets: CIFAR10 (Krizhevsky & Hinton, 2009), SVHN (Netzer et al., 2011) and Amazon Review Polarity (Zhang et al., 2015; He & McAuley, 2016). We first evaluate the ability of a small ResNet model (He et al., 2016a;b) to select a subset of points for much larger ResNet model on CIFAR10 and SVHN datasets, which have 50,000 and 604,388 training examples respectively. Using the small proxy model is both fast and accurate enough to select 50% and 60% subsets of SVHN and CIFAR10 while speeding up end-to-end wall-clock time by $1.8\times$ and $1.6\times$ respectively. We further demonstrate SVP's robustness by showing that it outperforms uniform subsampling for a variety of proxy models and metrics. Additionally, we provide an extreme example of this robustness using fastText (Joulin et al., 2016) as a proxy model for VDCNN29 (Conneau et al., 2017) on the Amazon Review Polarity dataset, which has 3,600,000 training examples. fastText is a fundamentally different architecture than VDCNN29 and over $100\times$ faster to train (e.g., 10 minutes instead of 17 hours), yet SVP can remove 20% of the data while maintaining VDCNN29's lower error.

Here, we compare the performance of SVP with uniform subsampling, as the other methods discussed in Section 2 require changes to the optimization procedure, and our method can be applied as a preprocessing step to improve their training time. We further note that core-set selection and class balancing techniques do not apply here as we do not have pre-designed features and the datasets are relatively balanced.

**Implementation details.** In our experiments, we first train the large target model on the full dataset for $n_t$ epochs and use it as the baseline. We then train the proxy model on the entire dataset for a smaller number $n_p \leq n_t$ of epochs (as specified in Table 1). The trained proxy model is then used to select subsets of uncertain points to train the target model for the same $n_t$ number of epochs. Throughout this paper we report the mean error and standard deviation of 3 runs for each combination of proxy, target, and subset size, reducing the impact of random variations. For task specific hyperparameters, please see section 6.1.

**Wall-clock time.** To demonstrate the efficiency of SVP, we compare the wall-clock training time of selection via proxy to training over the full dataset with the target in Figure 4. For CIFAR10, we are able to maintain the same predictive performance as training over the entire dataset with 60% of the data, leading to an average speed-up of $1.6\times$ over 3 runs. In more detail, we use an ensemble of 3 ResNet20 models, where each model is taken based on the best validation error after 50 epochs of training, and rank examples based on their entropy. The rankings from each individual model are combined through rank combination, where the new rank for each example is the sum of its rank from each ResNet20 model. With this new ranking, we take the top 60% of examples and train ResNet164 with preactivation from scratch for a full 181 epochs as in He et al. (2016b). The slowest ResNet20 model takes 6 minutes and 20 seconds to train, which is less than $1/30$ of the original training time for ResNet164. However, using the ranking from this small proxy model, we are able to remove 40% of the data and make up for the increase in overhead as demonstrated in Figure 4a.

Similarly, for SVHN, we are able to maintain the same predictive performance as training over the entire dataset with 50% of the data, leading to an average speed-up of $1.8\times$. Unlike CIFAR10, ensembling does not improve or harm data selection, so we rank points based on loss from only a single ResNet20 model after 10 epochs of training, which takes an average of 13 minutes and 4 seconds on a Titan V GPU. Using this ranking, we eliminate 50% of the examples and train ResNet152 on the remaining images for a full 50 epochs as in Huang et al. (2016). Including the time to train the proxy and make the selection, training ResNet152 to the same error level takes 2

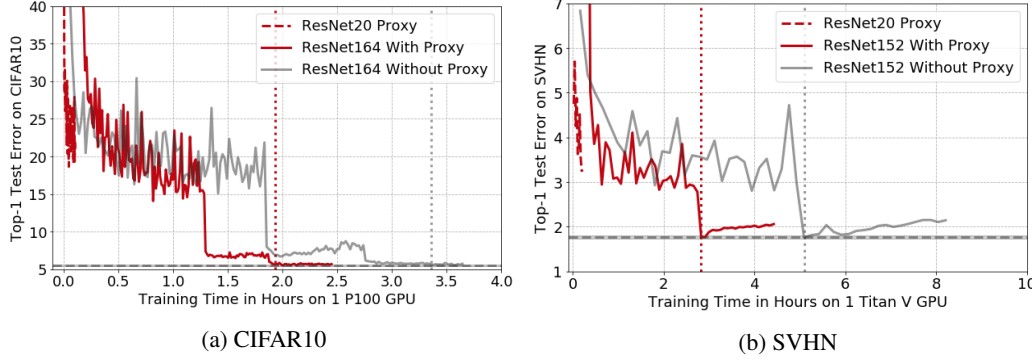

(a) CIFAR10                  (b) SVHN

Figure 4: Training curves of ResNet164 with pre-activation on CIFAR10 (left) and ResNet152 on SVHN (right) with and without data selection via proxy. The dashed gray line shows the average error of the target model across 3 runs. The dashed red line shows training the proxy model. The solid red line shows training the target model on a subset of images selected by the proxy. The solid gray shows training the target model on full dataset. The dotted red and gray lines show when each training curve gets within 1 standard deviation of the target model's error on the fully dataset. CIFAR10 and SVHN only require 60% and 50% of the data to maintain the same error level.

Table 1: Average Top-1 error and standard deviation for 3 runs of different proxy models across a range of subset sizes of the CIFAR10, SVHN, and Amazon Review Polarity datasets. '-' indicates that the SVP reached within 1 standard deviation of the average error of the target model trained over all of the dataset with a smaller subset size, adding more data does not result in significant improvements and performance plateaus.

| Dataset | Proxy Architecture | Metric | Epochs ($n_p$) | Fraction of Dataset 0.4 | 0.6 | 0.8 | 1.0 |
|---|---|---|---|---|---|---|---|
| CIFAR10 | 3xResNet20 | Entropy | 50 | $6.52 \pm 0.21$ | $5.46 \pm 0.06$ | - | - |
| CIFAR10 | 1xResNet20 | Entropy | 50 | $6.83 \pm 0.07$ | $5.61 \pm 0.09$ | - | - |
| CIFAR10 | 1xResNet20 | Entropy | 180 | $7.09 \pm 0.17$ | $5.71 \pm 0.22$ | $5.53 \pm 0.23$ | - |
| CIFAR10 | 1xResNet164 | Entropy | 181 | $7.83 \pm 0.32$ | $6.31 \pm 0.15$ | $5.68 \pm 0.25$ | $5.48 \pm 0.08$ |
| CIFAR10 | | | | $8.93 \pm 0.19$ | $6.87 \pm 0.16$ | $6.07 \pm 0.10$ | $5.52 \pm 0.12$ |
| SVHN | 1xResNet20 | Entropy | 10 | $1.87 \pm 0.03$ | $1.72 \pm 0.04$ | - | - |
| SVHN | 1xResNet20 | Entropy | 50 | $1.94 \pm 0.20$ | $1.86 \pm 0.05$ | $1.79 \pm 0.02$ | - |
| SVHN | | | | $2.27 \pm 0.06$ | $1.98 \pm 0.05$ | $1.88 \pm 0.04$ | $1.79 \pm 0.06$ |
| Amazon Review Polarity | 1xfastText | Entropy | 5 | $4.39 \pm 0.02$ | $4.23 \pm 0.02$ | $4.16 \pm 0.02$ | - |
| Amazon Review Polarity | | | | $4.89 \pm 0.03$ | $4.50 \pm 0.05$ | $4.28$ | $4.13 \pm 0.04$ |

hours and 50 minutes rather than 5 hours and 5 minutes when training over all of the data as shown in Figure 4b.

For Amazon Review Polarity, we are able to maintain the same predictive performance with VD-CNN29 as training over the entire dataset while removing 20% of the dataset using fastText as a proxy as shown in Table 1. In comparison to VDCNN29, which takes 16 hours and 40 minutes to train over the entire dataset on a Titan V GPU, fastText two orders of magnitude faster, taking less than 10 minutes on a laptop to train over the same data and compute output probabilities. This allows us to train VDCNN29 to the same error level in 13 hours and 30 minutes.

**Comparing different proxies.** Table 1 shows the impact of ensembling, partial training, and model architecture on selecting data from a proxy. Ensembling and partial training both improve performance, allowing a subset of 60% of the data to maintain the same predictive performance as the full dataset on CIFAR10. For ensembling, rank combination outperformed alternating selection (i.e., taking the most uncertain points from each model in a robin-round fashion). Without either ensembling or partial training, selection with ResNet20 degrades and requires 70% to 80% of the data instead. Surprisingly, using a fully trained ResNet164 with pre-activation performs worse than using ResNet20 with pre-activation to select examples for a separate ResNet164 with pre-activation. The favorable performance of smaller architectures and partial training might be a result of increased randomness and better coverage of the entire datasets.

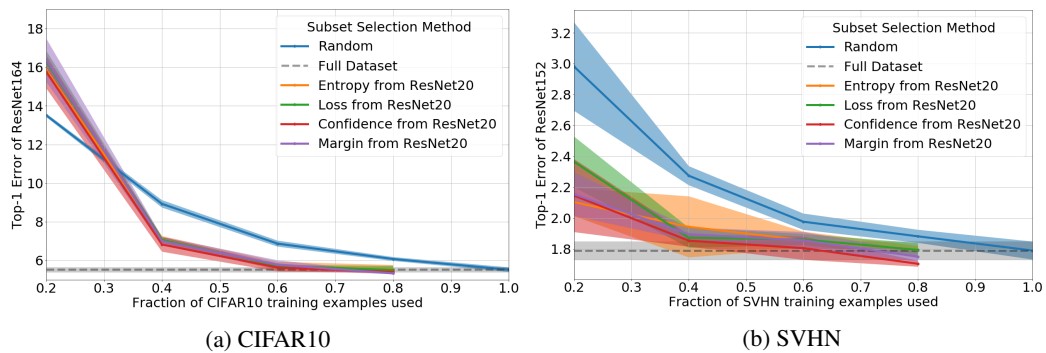

|  | (a) CIFAR10 | (b) SVHN |
|---|---|---|

Figure 5: Test error of ResNet164 with pre-activation on CIFAR10 (left) and ResNet152 on SVHN (right) with different uncertainty metrics. The dashed gray line is the mean error over 3 runs of the target model on all of the dataset. The markers on the solid lines show the mean error over 3 runs of different selection methods for a given subset size. The shaded area is $\pm 1$ standard deviation.

**Comparing uncertainty metrics.** We experimented with various ways to quantify uncertainty and select points. Figure 5 shows the impact of different uncertainty metrics for a pair of proxy and target model. For both CIFAR10 and SVHN, all of the metrics performed similarly. While the metrics had a more variance on SVHN than CIFAR10 as shown in Figure 5b, this is mostly due to the variability between runs of the proxy model than the metrics. Unlike the Pearson correlation between models as shown in Figure 3, the correlation between metrics of the same model is always above 0.96.

**Class imbalance.** To understand which examples were being selected, we looked at the class distribution for multiple runs of ResNet20 with pre-activation on CIFAR10 at different subset sizes and points during training as shown in Table 2. Very small subsets had high class imbalance, but as training continues or the subset size increases the class distribution becomes more balanced. We hypothesized that forcing the selected subset to be more balanced would improve performance and reduce the size of the subset needed to maintain predictive performance, but we found that balancing the subset by selecting the most uncertain example from each class in a round-robin fashion actually harmed performance slightly.

Table 2: Class distribution for 3 runs of ResNet20 at different points during training and subset sizes.

|  |  | Classes |  |  |  |  |  |  |  |  |  |
|---|---|---|---|---|---|---|---|---|---|---|---|
| **Epoch** | **Data** | plane | car | bird | cat | deer | dog | frog | horse | ship | truck |
| 50 | 0.2 | 7.55±1.79 | 4.95±1.49 | 15.64±3.30 | 17.54±4.77 | 14.93±5.78 | 12.67±1.54 | 9.18±2.51 | 8.14±1.62 | 4.36±1.17 | 5.04±1.01 |
| 50 | 0.4 | 8.72±1.56 | 5.84±1.90 | 13.84±2.07 | 16.32±2.50 | 13.04±3.42 | 13.62±0.58 | 8.70±2.12 | 7.71±1.49 | 6.08±1.97 | 6.13±1.08 |
| 50 | 0.6 | 9.73±1.27 | 6.58±1.74 | 12.61±1.29 | 14.38±0.89 | 12.16±1.88 | 12.70±0.87 | 9.22±1.79 | 8.07±1.16 | 7.55±1.78 | 7.00±1.20 |
| 50 | 0.8 | 10.46±0.78 | 7.72±1.26 | 11.31±0.59 | 12.14±0.17 | 11.23±0.75 | 11.50±0.69 | 9.80±1.11 | 8.78±0.73 | 8.88±1.23 | 8.18±1.26 |
| 100 | 0.2 | 10.32±0.68 | 4.76±0.21 | 13.19±1.24 | 18.71±4.91 | 9.32±1.68 | 16.34±0.07 | 8.15±1.26 | 8.41±0.92 | 5.67±1.77 | 5.15±0.82 |
| 100 | 0.4 | 11.62±0.34 | 5.66±0.15 | 11.61±0.64 | 16.24±2.53 | 9.57±1.42 | 14.69±0.03 | 8.67±0.98 | 8.66±0.67 | 6.84±1.56 | 6.43±0.63 |
| 100 | 0.6 | 11.82±0.08 | 6.78±0.09 | 10.77±0.34 | 13.78±1.14 | 9.76±1.01 | 13.00±0.06 | 9.29±0.75 | 9.08±0.60 | 8.07±1.28 | 7.65±0.33 |
| 100 | 0.8 | 11.41±0.04 | 8.27±0.30 | 10.09±0.16 | 11.72±0.44 | 9.72±0.73 | 11.52±0.09 | 9.70±0.51 | 9.61±0.42 | 9.11±0.85 | 8.86±0.16 |
| 180 | 0.2 | 10.79±1.01 | 5.48±0.22 | 11.67±1.06 | 18.35±0.15 | 9.53±1.29 | 14.99±0.42 | 7.88±1.14 | 8.04±0.31 | 6.37±0.45 | 6.92±0.60 |
| 180 | 0.4 | 11.08±0.82 | 6.98±0.28 | 10.71±0.60 | 15.36±0.16 | 9.76±0.87 | 13.20±0.47 | 8.58±0.80 | 8.76±0.21 | 7.65±0.35 | 7.94±0.66 |
| 180 | 0.6 | 10.91±0.46 | 8.53±0.47 | 10.14±0.41 | 13.13±0.13 | 9.64±0.55 | 11.98±0.32 | 9.05±0.56 | 9.19±0.32 | 8.61±0.52 | 8.81±0.40 |
| 180 | 0.8 | 10.61±0.30 | 10.04±0.44 | 9.71±0.24 | 11.30±0.11 | 9.53±0.42 | 11.06±0.24 | 9.25±0.30 | 9.55±0.28 | 9.41±0.40 | 9.54±0.27 |
| - | 1.0 | 10.00±0.00 | 10.00±0.00 | 10.00±0.00 | 10.00±0.00 | 10.00±0.00 | 10.00±0.00 | 10.00±0.00 | 10.00±0.00 | 10.00±0.00 | 10.00±0.00 |

## 5  CONCLUSION

In this work, we present *Select Via Proxy* (SVP), a novel approach to efficiently select a subset of training data to achieve faster training of deep learning models with no loss in predictive performance. Using this approach, we demonstrate that a small proxy model that is more than $30\times$ faster to train can select a subset of data to train a large architecture, while maintaining the predictive performance. On CIFAR10 and SVHN, the speed of training the proxy model leads to a $1.6\times$ and $1.8\times$ speed-up in end-to-end training time by selecting 60% and 50% of data respectively to train the target model on. Aside from filtering input examples, we do not change the training procedure of the target model, making our method a modular component to add to existing training pipelines.

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

# 6 APPENDIX

## 6.1 HYPERPARAMETERS

**CIFAR10.** We used ResNet164 with pre-activation from He et al. (2016b) as our large target model. Based on the trade-off between training time and error in Figure 2a, we chose ResNet20 with pre-activation as the proxy architecture, where the filters and layers were scaled down to match He et al. (2016a). To avoid extensive hyperparameter search, we followed the same training procedure, initialization, and hyperparameter as He et al. (2016b) with the exception of weight decay, which was set to 0.0005 and decreased the model's error under all conditions.

**SVHN.** We used ResNet152 and ResNet20 from He et al. (2016a) as the large target model and the small proxy model respectively. We followed the same training procedure, initialization, and hyperparameters from Huang et al. (2016).

**Amazon Review Polarity.** we used fastText as the proxy model and VDCNN29 as the target model and followed the same training procedure, initialization, and hyperparameters from Joulin et al. (2016) for fastText and from Conneau et al. (2017) for VDCNN29.

