# OpenReview forum: "Select Via Proxy: Efficient Data Selection For Training Deep Networks"
_ICLR.cc/2019/Conference_

### Official Review · AnonReviewer1 · 2018-10-31
**meaningful work, but lacks more supporting evidences**

**Rating:** 5
**Confidence:** 4

**Review:**

This paper studies a very simple and intuitive method to boost the training speed of deep neural networks. The authors first train some light weighted proxy models, using these models to rank the data according to its uncertainty, and then pick the most uncertain subset to train the final model. Experiments on CIFAR10/SVHN/Amazon Review Polarity demonstrates the effectiveness.

In general, I think the authors did a decent job in showing that such a simple idea could surprisingly work well to boost NN training. I believe it will inspire future works on speeding up NN training. However, to form a solid ICLR publication, plenty of future works need to be done.

1)	I will not be fully convinced if an idea aiming to speed up, is only verified on small scale dataset (e.g., CIFAR10). It will be much better if there are large scale experiments conducted such as on ImageNet and WMT neural machine translation.

2)	Please well position some related works. First, it would be more interesting and informative if some baselines in section 2 (especially those in “Optimization and Importance Sampling’), are compared with. Second, there are important related works omitted such as L2T [1], which also talks/shows the possibly of using partial training data to achieve speed up.

3)	Some writing issues: it would be better to *clearly* demonstrate the final accuracy of different models (i.e. ResNet 164 trained on whole data and selected subset), such as putting them into a table, but not merely showing them vaguely in the curves and text. I’m also note sure about the meaning of `epoch’ in Table 1: does it mean how many epochs the proxy model is trained? If so, I can hardly get the intuition of why smaller epochs works better. I noted a conjecture raised by the authors in the last sentence of paragraph “comparing different proxies”. However, I cannot catch the exact meaning.

[1] Fan, Y., Tian, F., Qin, T., Li, X. Y., & Liu, T. Y. Learning to Teach. ICLR 2018

---

> ### Author Response · Authors · 2018-11-27
> **Experiments in progress and clarifications about figures**
>
> Thank you for your thoughtful review and suggestions. Here are our responses:
>
> # large-scale experiments
> We are currently running experiments on ImageNet, but the results will not be ready before the response deadline.
>
> # Comparison against baselines in section 2
> We agree that more comparison against existing methods such as importance sampling would be valuable. we aimed to compare against “Not All Samples Are Created Equal: Deep Learning with Importance Sampling” from Katharopoulos & Fleuret (2018) as it represents the most recent published work in the area. Unfortunately, we were unable to complete the experiments before the response deadline.
>
> # Learning to teach (L2T)
> We agree learning to teach is relevant and included it in the related work section.
>
> # Final accuracy of different models in a table
> We believe Table 1 should address this concern, and we changed the structure to make it more clear. The most important data from Figure 4 and 5 is captured in the table. We could add the additional metrics from Figure 5, but the main point of that figure is to show that all of the metrics perform about the same, which would just add redundant rows to the table.
>
> # Smaller number of epochs
> Great question. Yes, "epoch" in Table 1 means how many epochs the proxy model is trained. We have preliminary results that suggest the diversity of the subset is an important factor in maintaining quality. Looking at the CDF of entropy on CIFAR10, for example, shows that only around 20% of points have relatively high entropy and that entropy quickly decays after the first 20%. However, the target model is only able to match the same level of accuracy with a larger subset as shown in Table 1. This suggests that the subset needs to be sufficiently representative in addition to containing the most difficult. We hypothesize that the higher error of smaller architectures and partial training might result in increased randomness, which could improve the representativeness of the resulting subsets.

---

### Official Review · AnonReviewer2 · 2018-11-02
**Interesting approach to data selection, but needs comparative experiments**

**Rating:** 4
**Confidence:** 2

**Review:**

# Summary
The paper presents a method for identifying and selecting the most informative subset of the training dataset in order to reduce training time while maintaining test accuracy. The method consists of training a proxy model that is smaller and has been trained for fewer epochs, and which can optionally be ensembled. Experiments show promising results, indicating that some datasets can be reduced to half the size without impacting model performance.

# Quality
The paper appears sound and of good quality. Background literature is cited and the proposed method is discussed in sufficient detail.

I would, however, like to see some additional comparative experiments. All experiments are constructed to show that the method can indeed achieve accuracy comparable to the full model but with a smaller training set. I would like to see how it compares to existing strategies -- are there any reason to pick this method over existing ones?
Since the last sentence in section 2 states that the proposed method is orthogonal to previous subsampling techniques, and therefore can be combined with any of them, it would be interesting to see how SVP compares to these and whether a combination of, say, SVP and importance sampling will in fact achieve better performance than the importance sampling on its own.
Additionally, given the model's high resemblance to active learning, it would be interesting to see it compared to some prominent active learning methods.

# Clarity
The paper reads quite well. I particularly like the paragraph headlines, which makes it easy to get an overview of the paper.

The figures are generally nice and readable, except for figure 3, which I don't understand. Maybe I am missing it, but I can't find an explanation for what the rows and columns indicate, and the labels themselves should also be increased in size.

# Originality
I do not find the paper particularly novel. To me, the proposed method seems to be a variant of active learning, not orthogonal to this as it is claimed in section 2. The choice of surrogate model and uncertainty metric might be new, but the method itself boils down to uncertainty sampling, a well-known strategy in active learning.
However, I am happy to change my mind if the authors can explain to me exactly how their method differs from active learning.

# Significance
While techniques for speeding up training without sacrificing performance are, of course, always interesting, I find the proposed method to be rather incremental and not significant enough for ICLR. It would be better suited as a workshop paper.

# Other notes
In the last paragraph of section 1, you write that "Our proposed framework is robust to the choice of proxy model architecture." I am not sure what you mean by this. Do you mean that one can choose any model as the proxy (which is clearly correct) or do you mean that the method is "proxy agnostic" in the sense that any proxy model will work better than no proxy? If the latter is the case, I would like some arguments for this. Also, if the method is indeed proxy agnostic, it should be possible to remove the proxy completely and select the data in some other way.

---

> ### Author Response · Authors · 2018-11-27
> **Clarifications about novelty and impact**
>
> Thank you for your thoughtful review and suggestions. Here are our responses:
>
> # SVP and importance sampling
> We agree that more comparison against existing methods such as importance sampling would be valuable. We aimed to compare against “Not All Samples Are Created Equal: Deep Learning with Importance Sampling” from Katharopoulos & Fleuret (2018) as it represents the most recent published work in the area. Unfortunately, we were unable to complete the experiments before the response deadline.
>
> # Active learning, originality, and significance
> We agree that we leverage uncertainty sampling (Lewis & Gale, 1994) from active learning to select the points with highest informativeness. However, in active learning a model is generally trained to select the next point (Settles, 2012) or batch (Sener & Savarese, 2018), which is efficient in terms of labels, but often computationally expensive. While this can be effective when deciding which data to acquire labels for from an expensive labeler (e.g. a human), the computational cost is too high to accelerate training over an existing large labeled dataset. Using a proxy reduces the cost of selection by up to a 100x for Amazon Review Polarity or 30x for CIFAR10. This is such a substantial improvement that uncertainty sampling can now be extended to reduce computational costs of training in addition to labeling costs. To clarify this point, we added more detail in the introduction.
>
> # Figure 3: model correlation
> We increased the size of the labels and improved the figure caption. The key takeaway is that ensembling multiple small models together through rank combination improves our approximation of the large model’s uncertainty as shown by the increase in correlation ranking.
>
> # Robustness to the choice of proxy model architecture
> By “robust to the choice of proxy model architecture” we mean that while we discussed different steps in Section 3.1 to create the proxy model, and explored various uncertainty measures in Section 3.2 to select data points via proxy, our approach allows for a wide range of configurations. The proxy is important but it is easy to find one that is good enough in practice and doesn’t require extensive hyperparameter tuning.

---

### Official Review · AnonReviewer3 · 2018-11-02

**Rating:** 4
**Confidence:** 4

**Review:**

General:
The paper proposed an algorithm named Select Via Proxy(SVP), which can be used for data sampling. The idea is simple and straightforward: 1) use a proxy model to get decision boundary 2) train the large target model on the data points close to the decision boundary.

Strength:
1. Roughly this is a well-written paper. The main idea is quite clear to me.
2. Empirical validation of the experiments looks good. The results show that SVP help reduce the training time with ResNet. The author(s) also showed the influence of different quantifying uncertainty methods.

Possible Improvements:
1. In Related Work, several previous works were mentioned. Although the author(s) claimed that SVP can be combined with them, it's better to show the performance of SVP compared with them. This would show the significance of the work.
2. In the experiments, I was hoping to see how well SVP works on ImageNet. The problem is that: For ResNet152 and ResNet164, they are relatively too deep on such small data sets. Since the dimension of the data points(images) is not high, SVP can easily catch a reasonable decision boundary with a smaller model. I am almost sure ResNet20 is good enough to do this. I am more concerned about the situation where the capacity of the model is challenged by the size of the dataset. e.g. The data sets of autonomous driving are usually extremely large and even very deep models cannot be fully trained on that.
3. The data points close to the decision boundary can be considered as tough data points, whose features might be hard to be caught by the model. If training model only on these data points, the trained model may just memorize tough data points and not learn the other data points from the data set. One solution is that, while training on tough data points, the model should also be trained on a small portion of well-learned data points. I don't think training only on the points close to the decision boundary is enough and was more expecting to see some discussion about this in the paper.

Conclusion:
My two biggest concerns are: 1) The algorithm is not tested on large data seta 2) The algorithm is not tested with the models of limited capacity. As a conclusion, I tend to vote for rejection.

---

> ### Author Response · Authors · 2018-11-27
> **Experiments in progress**
>
> Thank you for your thoughtful review and suggestions. Here are our responses:
>
> 1) We agree that it would be valuable to provide more comparison against related work. The work in the “Optimization and Importance Sampling” section shares the closest relation to our approach by improving convergence and in some cases training time. In particular, we aimed to compare against “Not All Samples Are Created Equal: Deep Learning with Importance Sampling” from Katharopoulos & Fleuret (2018) as it represents the most recent published work in the area. Unfortunately, we were unable to complete the experiments before the response deadline.
>
> 2) We are currently running experiments on ImageNet, but the results will not be ready before the response deadline. However, for both SVHN and CIFAR10, the larger model improves accuracy significantly over the proxy. For CIFAR10, there is a 2.5% difference in absolute error (47% relative error) between ResNet20 and ResNet164 . With partial training, the difference in absolute error between the ResNet20 and ResNet164 is ~12%. Interestingly, despite the limited capacity of the proxy models, our approach still selects a subset that maintains accuracy and performs much better than random. In fact, for CIFAR10, using ResNet20 as the select proxy performs better than using ResNet164 (the target model) for selection as shown in Table 1.
>
> 3) This is a great observation. We have preliminary results that suggest the diversity of the subset is an important factor in maintaining quality. Looking at the CDF of entropy on CIFAR10, for example, shows that only around 20% of points have relatively high entropy and that entropy quickly decays after the first 20%. However, the target model is only able to match the same level of accuracy with a larger subset as shown in Table 1. This suggests that the subset needs to be sufficiently representative in addition to containing the most difficult. Fortunately, we hypothesize that the proxy model can also give us an efficient way to calculate the representativeness of each example as well as uncertainty, allowing us to algorithmically construct such as subset. We are already looking into this but couldn’t complete the necessary experiments while also attempting to address your feedback above. However, we can include CDFs of entropy and some preliminary discussion of this point.

---

### Public Comment · (anonymous) · 2018-10-10
**question about your uncertainty quantifying approach**

You do provide an efficient and simply idea (finding the subset of the original training data to achieve comparable performance with less complexity)  for many time-consuming training tasks in CV and NLP fields .

However, your formulas in quantifying the uncertainty merely take the classification probability into consideration, which can not be an general data selection model , have you thought that before?  anyway  no offense , the novelty of this paper is not enough~  hope you do not mind my one-sided comment : )

---

> ### Author Response · Authors · 2018-10-12
> **the surprising result is that simple metrics from small proxy models are so effective.**
>
> Thank you for your comment. Our key observation is that the behavior of a smaller model is a useful and efficient proxy for training data selection.  After evaluating entropy, loss, confidence, and margin as shown in Figure 5, we found limited differences in performance. Of course, the use of uncertainty as a quality metric is just one option among many. We are already looking at additional formulations that use intermediate activations and combine uncertainty sampling with other metrics that focus on diversity. Our preliminary results show improved performance for small fractions of the data. We plan to include these results in the final version of the paper. However, for us, the surprising result is that simple metrics from small proxy models are so effective.

---

### Comment · Area_Chair1 · 2018-12-05
**why ever train for more than one epoch on the reduced dataset?**

I don't see why it would be useful to make multiple passes over the data used for training the target model (the selected subset of the training set). If you want to get a speedup in training, it should be strictly better to use fresh data and never look at any data point more than once. Wouldn't it be better to conduct experiments with one pass over the subsampled dataset to see how useful the algorithm is in this regime?

---

> ### Author Response · Authors · 2018-12-10
> **Popular benchmarks make multiple passes for higher accuracy and SVP with shorter schedules**
>
> Great question! Our method is generic and could be applied to training procedures that only make a single pass over the data. Unfortunately, we did not have access to a large task that is solved in a single pass over the dataset. To the best of our knowledge, all of the popular benchmarked datasets make multiple passes (epochs) over the dataset.
>
> However, we did consider a similar issue about whether more epochs over a small dataset might be the same as fewer epochs over a large dataset. Based on experimental results, condensing the learning rate schedule (i.e., fewer epochs) and using the full dataset hurts accuracy and can't achieve the same speed-up on CIFAR10. For SVHN, we can substantially condense the schedule from Huang et al. (2016), but SVP stacks with the condensed schedule, giving an additional speed-up (i.e. ~1.8x). We can include these additional results in the appendix of the final draft of the paper.
>
> Also, there is initial theoretical work that shows that one pass over the data isn't enough for hard problems:
>
> https://papers.nips.cc/paper/8034-statistical-optimality-of-stochastic-gradient-descent-on-hard-learning-problems-through-multiple-passes

---

### Meta-Review · Area_Chair1 · 2018-12-13
**Interesting task, but clear consensus among reviewers to reject this paper based on limited experiments**

**Confidence:** 5
**Recommendation:** Reject

**Metareview:**

There reviewers unanimously recommend rejecting this paper and, although I believe the submission is close to something that should be accepted, I concur with their recommendation.

This paper should be improved and published elsewhere, but the improvements needed are too extensive to justify accepting it in this conference. I believe the authors are studying a very promising algorithm and it is irrelevant that the algorithm is a relatively obvious one. Ideally, the contribution would be a clear experimental investigation of the utility of this algorithm in realistic conditions. Unfortunately, the existing experiments are not quite there.

I agree with reviewer 2 that the method is not particularly novel. However, I disagree that this is a problem, so it was not a factor in my decision. Novelty can be overrated and it would be fine if the experiments were sufficiently insightful and comprehensive.

I believe experiments that train for a single epoch on the reduced dataset are absolutely essential in order to understand the potential usefulness of the algorithm. Although it would of course be better, I do not think it is necessary to find datasets traditionally trained in a single pass. You can do single epoch training on other datasets even though it will likely degrade the final validation error reached. This is the type of small scale experiment the paper should include, additional apples-to-apples baselines just need to be added. Also, there are many large language modeling datasets where it is reasonable to make only a single pass over the training set. The goal should be to simulate, as closely as is possible, the sort of conditions that would actually justify using the algorithm in practice.

Another issue with the experimental protocol is that, when claiming a potential speedup, one must tune the baseline to get a particular result in the fewest steps. Most baselines get tuned to produce the best final validation error given a fixed number of steps. But when studying training speed, we should fix a competitive goal error rate and then tune for speed. Careful attention to these experimental protocol issues would be important.